

# *Saccharibacteria (TM7)*, but not other bacterial taxa, are associated with childhood caries regardless of age in a South China population

Yang You[1], Meixiang Yin[1,2], Xiao Zheng[1], Qiuying Liang[1], Hui Zhang[1], Bu-Ling Wu[3] and Wenan Xu[1]

[1] Department of Pediatric Dentistry, Shenzhen Stomatology Hospital (Pingshan), Southern Medical University, ShenZhen, GuangDong, China
[2] Department of Stomatology, Shenzhen Samii Medical Center, ShenZhen, GuangDong, China
[3] Department of Endodontics, Shenzhen Stomatology Hospital (Pingshan), Southern Medical University, ShenZhen, GuangDong, China

Corresponding authors
Bu-Ling Wu, bulingwu@smu.edu.cn
Wenan Xu, xu_wenan@smu.edu.cn

## ABSTRACT

**Background:** Human microbiome dysbiosis is related to various human diseases, and identifying robust and consistent biomarkers that apply in different populations is a key challenge. This challenge arises when identifying key microbial markers of childhood caries.

**Methods:** We analyzed unstimulated saliva and supragingival plaque samples from children of different ages and sexes, performed 16S rRNA gene sequencing, and sought to identify whether consistent markers exist among subpopulations by using a multivariate linear regression model.

**Results:** We found that *Acinetobacter* and *Clostridiales* bacterial taxa were associated with caries in plaque and saliva, respectively, while *Firmicutes* and *Clostridia* were found in plaque isolated from children of different ages in preschool and school. These identified bacterial markers largely differ between different populations, leaving only *Saccharibacteria* as a significant caries-associated phylum in children. *Saccharibacteria* is a newly identified phylum, and our taxonomic assignment database could not be used to identify its specific genus.

**Conclusion:** Our data indicated that, in a South China population, oral microbial signatures for dental caries show age and sex differences, but *Saccharibacteria* might be a consistent signal and worth further investigation, considering the lack of research on this microbe.

## INTRODUCTION

Childhood caries is the most common disease in children, affecting 0.5 billion children worldwide and leading to a heavy medical and economic burden (*Global oral health status report: towards universal health coverage for oral health by 2030*, WHO). Dental caries not only threaten the oral health of children but also affect the physical and mental health of children in severe cases (*Jeffrey, 2016*). According to the 4th National Oral Health Survey

in mainland China (*Wang, 2018*), the prevalence of dental caries in children was 50.8%, 63.6% and 71.9% for 3-, 4- and 5-year-olds, respectively; the mean number of decayed, missing, or filled teeth (dmft) at 5 years old was 4.24, and the untreated rate was 96.0%. There is no doubt that the oral microbiota is the key pathogenic factor leading to dental caries, and many factors affect its development. The human oral cavity hosts a highly diverse bacterial community; however, only approximately half of oral microbes can be cultured (*Paster et al., 2006*). Furthermore, children's transition from primary dentition to early mixed dentition begins when the children are 3 to 8 years old. The experience of caries in primary teeth can lead to a high risk of decay in permanent dentition (*Zou et al., 2022*). This critical period determines the oral health status in adulthood and is also a key period for the establishment of oral health concepts and behaviors. However, the flora was significantly different between individuals or in different regions of the same individual (*Mark Welch, Ramírez-Puebla & Borisy, 2020*), and different key bacteria have been observed in various studies, with the influencing factors behind the bacteria being of interest. Therefore, to study oral microbiota, it is necessary to understand their influencing factors.

In microbiome research, population association analysis is usually used to obtain key bacteria, and then the causal relationship and interaction mechanism between them and diseases are explored. A balanced microbiota is the foundation of oral health, as dysbiosis causes tooth decay (*Lamont, Koo & Hajishengallis, 2018*). For the past few decades, acid-producing bacteria represented by *Streptococcus* mutans have been considered the main pathogenic bacteria involved in dental caries (*Palmer et al., 2010*), and most preventive measures and risk assessment methods against caries target *Streptococcus mutans* (*Cugini et al., 2019*; *Zhang et al., 2021*). In recent years, high-throughput sequencing has been used to study caries-associated microorganisms, and the sequencing results of the key oral bacteria for dental caries are often inconsistent. It has been found that *S. mutans* is not the dominant bacterium in plaque biofilm, as it is present at a low level in some caries patients and can also be detected in caries-free individuals (*Dinis et al., 2022*). Thus, other bacteria must be involved in the process of caries development. However, a growing body of evidence supports the view that *S. mutans* is not a single causative factor for caries (*Kleinberg, 2002*; *Simón-Soro & Mira, 2015*). In addition to *S. mutans*, other bacterial genera, including *Lactobacillus*, *Actinomyces*, and *Veillonella*, are thought to be involved in the occurrence and development of dental caries (*Becker et al., 2002*; *Aas et al., 2008*). Some bacteria, such as *Veillonella*, have been found to exhibit high relative abundance at all stages of dental caries (*Aas et al., 2008*), and they can be involved in acid production at high glucose levels (*Bradshaw & Marsh, 1998*).

The relationship between bacterial flora and dental caries is complicated, and studies indicate that the population characteristics of key oral bacteria for dental caries remain unclear. Many studies have shown that dietary habits (*Mahmoud et al., 2022*), oral hygiene behaviors (*Finlayson et al., 2019*), genetics (*Valles-Colomer et al., 2023*), and the environment (*Shaw et al., 2017*) all affect children's oral microbiota. Therefore, it is increasingly important to identify the key bacteria that can be used as targets for pathological analysis or targeted drug therapy to avoid the side effects of prolonged

therapies with broad-spectrum drugs, both on the oral microbiota and at a systemic level (*Mahendra et al., 2021*). For children, the situation becomes more complicated due to tooth replacement. One possibility is that the key bacteria in children differ based on age, dentition and sex.

The relationship between dental caries and bacterial flora may be a new target for dental caries risk assessment and intervention. Most previous studies involving dental caries in children have focused on the comparison of oral microbes between children with and without caries (*Jiang et al., 2016*; *Kahharova et al., 2020*; *Qudeimat et al., 2021*). However, there are few studies on the correlation between children with different caries severity. Dental plaque is the driving factor of caries, but the primary medium in the oral environment is saliva. It is often used as a sampling site for microbial detection because of the convenience of this analysis. Therefore, this experiment was designed to collect supragingival plaque and unstimulated saliva from children.

In this cross-sectional study, we explored the relationship between caries severity and the oral microbiota. In total, 102 children with caries were included in this study, and high-throughput sequencing of supragingival plaque and unstimulated saliva samples from children with caries was used to analyze similarities and differences in key caries-related microbes according to the children's age and sex.

## MATERIALS AND METHODS

### Participants

The study was conducted in accordance with the Declaration of Helsinki and was approved by the Ethics Committee of Shenzhen Stomatology Hospital (Pingshan) of Southern Medical University (202201A). Study participants were recruited from the Department of Pediatric Dentistry in our hospital from March 2022 to September 2022 during their first visit. Patients who were scheduled to see a doctor at 9–11 am were selected for this study, and 127 children aged 3 to 8 years old were initially enrolled. After sample sequencing screening, donors with unqualified PCR and samples with less than 30,000 reads sequenced were excluded. Finally, 102 children remained. All parents or other legal guardians of the participating children signed the informed consent form. A telephone call was made the day before the first visit, and preliminary communication was made in accordance with the inclusion and exclusion criteria.

### Criteria

Inclusion criteria were as follows:

a. children aged 3–8 years old

b. children with caries (ICDAS of any tooth surface ≥3) who have never been treated before

c. children in the mixed dentition stage had no loss of deciduous teeth within 1 month

d. children in the mixed dentition stage in whom the deciduous molars had not been replaced and without caries in the new permanent teeth

e. lack of systemic diseases or congenital diseases

f. no obvious active bacterial or viral infection

g. informed consent was obtained from guardians.

The exclusion criteria were as follows:

a. use of antibiotics, probiotics, synbiotics or fluoride in the 3 months prior to the study

b. developmental diseases of teeth

c. extrinsic black tooth stain

d. completion of orthodontic treatment within 6 months or undergoing orthodontic treatment

e. mucosa or salivary system disease

## Sample collection

According to the "Oral Microbiome: Methods and Protocols" (*Hussein, 2021*), on the day before the clinic visit, we communicated with participants' guardians about their intention to participate by telephone. Participants were required to refrain from tooth brushing the night before and the morning on the day they visited and to fast and not drink within 2 h before arriving at the hospital. Supragingival plaque was assessed using a sterile dental excavator and sterile swabs to collect the plaque on the surface of teeth. The samples were placed into a 1.5 ml sterile microcentrifuge tube (FCT217; Beyotime, Jiangsu, China) containing 0.8 ml PBS (C0221A; Beyotime, Jiangsu, China).

For the collection of unstimulated saliva, we allowed the patients to place a sterile cotton roll (Ruixue Medical Supplies, RX58, Shijiazhuang City, Hebei Province, China) under their tongue without moving, and then advised them to gently closed their mouths, lower their heads, rest, and avoid swallowing for 1 min; we then removed the cotton roll from the mouth with a dental tweezer and placed it into a 5 ml sterile microcentrifuge tube (FTUB020; Beyotime, Jiangsu, China). All the tubes were quickly placed in the freezer box of a foam incubator and stored in the refrigerator at −80 °C within 2 h until further processing.

## Clinical examination

Oral examinations were performed by an experienced dentist. Community periodontal index (CPI) probes and mirrors under optimal light assisted by air water syringe conditions were available for all dental examinations. Oral examination of the patients was performed using the International Caries Detection and Assessment System (ICDAS) criteria (*Ismail et al., 2007*). We polished the teeth with a rubber cup first and then moistened and dried the teeth for 5 s. The visual examination was coded 0–6.

## 16S rRNA amplicon sequencing

DNA was extracted using Mabio Bacterial DNA Extraction Mini Kits (Mabio Biotechnologies. Co., Ltd., Guangzhou, China) for the corresponding sample. The concentration and purity were measured using a NanoDrop One (Thermo Fisher Scientific, Waltham, MA, USA) to test the concentration and purification of the final DNA.

Primers 338F (5′-ACTCCTACGGGAGGCAGCA-3′) and 806R (5′-GGACTACHVGGGTWTCTAAT-3′) were used to amplify the V3-V4 hypervariable regions of the bacterial 16S rRNA gene using the PCR instrument Bio-Rad S1000 (Bio-Rad Laboratory, Hercules, CA, USA).

Data were collected as previously described in *Chen et al. (2023)*. Following the manufacturer's instructions, sequencing libraries were created using the NEBNext® Ultra™ II DNA Library Prep Kit for Illumina® (New England Biolabs, Ipswich, MA, USA), and index codes were added. The Qubit@ 2.0 Fluorometer (Thermo Fisher Scientific, MA, USA) was used to evaluate the library's quality. On the Illumina Nova6000 platform, the library was finally sequenced, and 250 bp paired-end reads were produced.

## Statistical analysis

After sequencing, 30,000 reads were set for data extraction, and the sample sequences were randomly selected to establish a uniform data volume to ensure the uniformity of the sample sequences. Bioinformatics analysis was conducted using QIIME2. DADA2 was used for sequence filtering, dereplication, sample inference, chimera identification, and merging of paired-end reads. All sequences were divided into amplicon sequence variants (ASVs) to reduce redundant computation during analysis.

Microsoft Excel 2016 was used for two-person data entry, and then SPSS 25.0 software was used to analyze the demographic data and experimental data. The chi-square test was used to analyze count data, such as sex and caries prevalence. Measurement data, such as age, BMI, and mean number of caries, were expressed as the mean ± standard deviation, and a $t$-test was used for analysis. For sequencing data, a Kruskal–Wallis nonparametric ANOVA calculated at the feature level was used to determine taxonomic group differences. For all statistical analyses, a $p$ value of less than 0.05 or a $q$ value of less than 0.25 indicated a significant difference.

## RESULTS

### Study demographics and sequencing results

The analyses included 102 children from 3 to 8 years of age that were caries-affected pediatric participants who lived in PingShan District, ShenZhen, with a mean ± SD age of 79.9 ± 21.47 months (Table 1). Each child contributed two samples from supragingival dental plaque and unstimulated saliva. Thus, a total of 204 samples were collected from these children. There were no significant differences in age, BMI, Hellman stage, number of decayed deciduous teeth or decayed deciduous teeth surface between the male and female groups.

A total of 24,274,860 high-quality 16S rRNA sequences were produced, with an average of 95,948 reads per sample. The number of ASVs was 10798. There were 1,849 ASVs in common between the two groups, 5,110 ASVs in the supragingival plaque group and 3,839 ASVs in the saliva group (Fig. S1). The rarefaction curves of all the samples trended flat at the end (Fig. S2). Sequences from these samples were classified into 28 phyla, 129 classes, 232 orders, 342 families, 467 genera, and 579 species.

**Table 1 Demographic and clinical characteristics of the individuals.**

|  | Total (*n* = 102) | Male (*n* = 61, 59.8%) | Female (*n* = 41, 40.2%) | *p*-value[*] |
|---|---|---|---|---|
| Age month (mean ± SD) | 79.91 ± 21.47 | 80.75 ± 20.57 | 78.66 ± 22.95 | 0.562[a] |
| BMI | 15.62 ± 2.46 | 15.59 ± 2.36 | 15.66 ± 2.61 | 0.768[a] |
| Age |  |  |  | 0.711[b] |
| Preschool | 45 (44.1%) | 26 | 19 |  |
| Primary school | 57 (55.9%) | 35 | 22 |  |
| Hellman stage |  |  |  | 0.187[b] |
| IIA | 36 | 24 | 12 |  |
| IIC | 8 | 2 | 6 |  |
| IIIA | 55 | 33 | 22 |  |
| IIIB | 3 | 2 | 1 |  |
| dt |  |  |  | 0.688[b] |
| 1–4 | 28 (27.45%) | 15 | 13 |  |
| 5–8 | 43 (42.16%) | 26 | 17 |  |
| 9–20 | 31 (30.39%) | 20 | 11 |  |
| ds | 13.18 ± 10.44 | 13.85 ± 10.89 | 12.17 ± 9.79 | 0.534[a] |

Notes:
Abbreviations: BMI, body mass index; dt, decayed deciduous teeth; ds, decayed deciduous teeth surface.
[a] *t*-test.
[b] Chi-Square.
[*] *p*-value Comparison between sexes.

## Microbiome profile in children with caries

We assessed alpha diversity, which is indicative of species abundance within each group. There were significant differences between the plaque and saliva groups in terms of richness (observed OTUs, Kruskal−Wallis, $p < 0.001$; Fig. 1A), alpha diversity (PD whole tree, Shannon index, Kruskal−Wallis, $p < 0.001$; Fig. 1A), and beta diversity (Bray−Curtis, PERMANOVA, $p = 0.001$; Fig. 1B). In saliva, the Shannon index values, which represent the richness and evenness of the distribution of the microbial community, were significantly lower than those in the plaque site ($p < 0.001$). Our findings suggest that the bacterial community in the plaque is more evenly distributed than that in saliva.

The children's bacterial community was primarily composed of six phyla, namely *Firmicutes*, *Proteobacteria*, *Actinobacteriota*, *Fusobacteriota*, *Bacteroidetes*, and *Saccharibacteria* (formerly known as *TM7*) (Fig. 1C). Figure 1D depicts the top 10 prevalent bacterial genera. In both saliva and plaques groups, *Streptococcus* (P:13,51%; S:37.49%) was the most dominant genus, followed by *Neisseria* (P:14.47%; S:6.14%), *Leptotrichia* (P:9.88%; S:5.12%), *Haemophilus* (P:4.23%; S:7.30%), *Fusobacterium* (P:7.80%; S:2.63%), *Corynebacterium* (P:6.46%; S:2.45%), *Actinomyces* (P:6.46%; S:2.42%), *Rothia* (P:3.70%; S:2.37%), *Veillonella* (P:4.87%; S:1.43%), and *Prevotella* (P:4.23%; S:1.87%).

Following this, we utilized LEfSe to identify the taxa that substantially varied across the samples for plaque and saliva, as depicted in Fig. 1E. As shown in supplementary, at the phylum level (Fig. S3A), *Firmicutes* was enriched in the saliva group, while *Proteobacteria*,

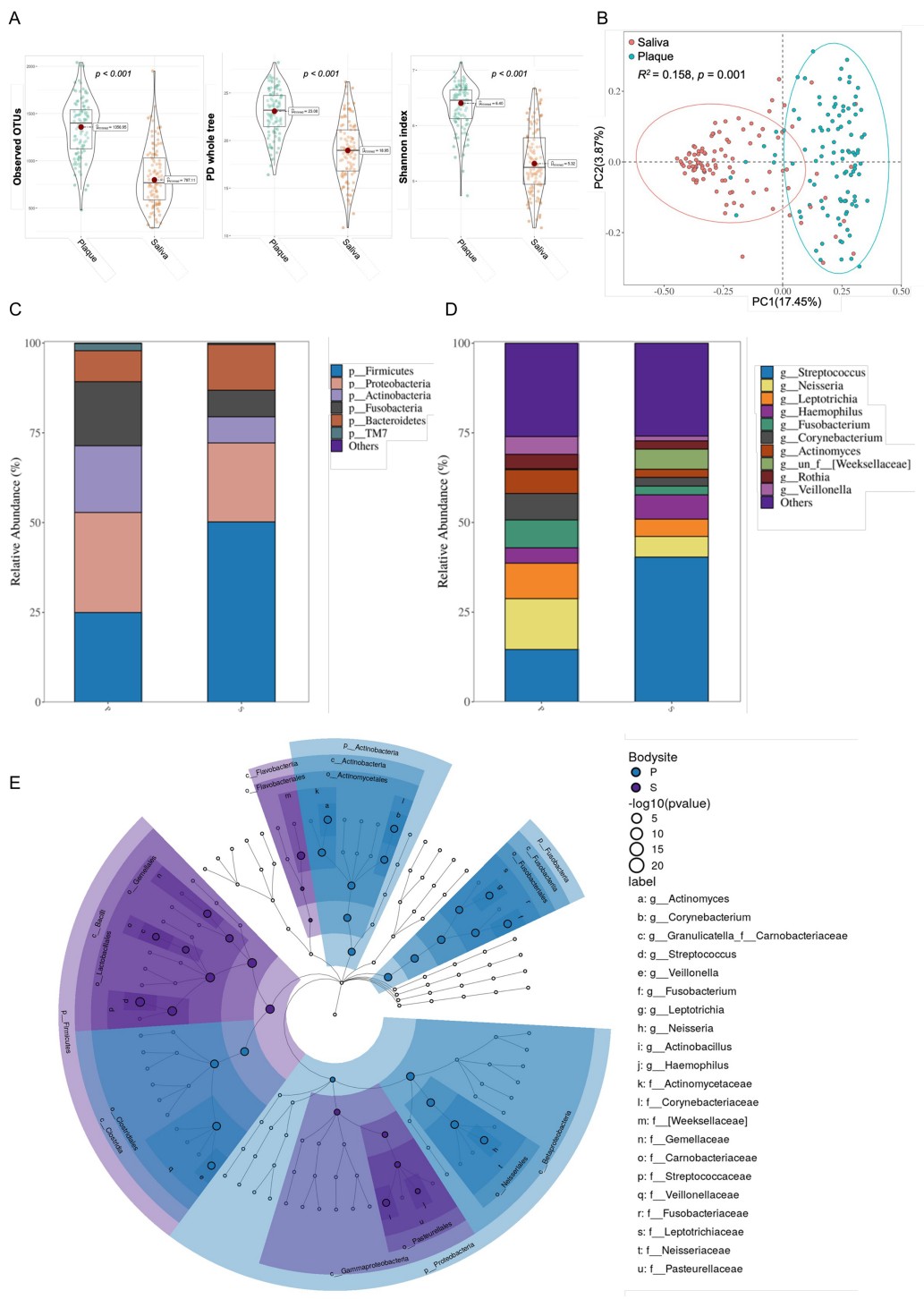

**Figure 1 Microbiota composition in the plaque (P) and saliva (S) groups.** (A) Alpha diversity of the plaque and salivary microbiome among groups. Comparison of observed OTUs, PD whole tree and Shannon index; (B) A principle coordinate analysis (PCOA) plot was generated using ASV metrics based on beta diversity (Bray–Curtis index) for different groups. PERMANOVA was used for these statistical comparisons between groups; (C and D) mean relative abundances of predominant phyla (C) and genera (D) in the two groups are displayed; (E) the linear discriminant analysis (LDA) effect size (LEfSe) method was performed to compare taxa between the plaque and saliva groups. The colored nodes from the inner to outer circles represent the most abundant taxa from the phylum to genus level.

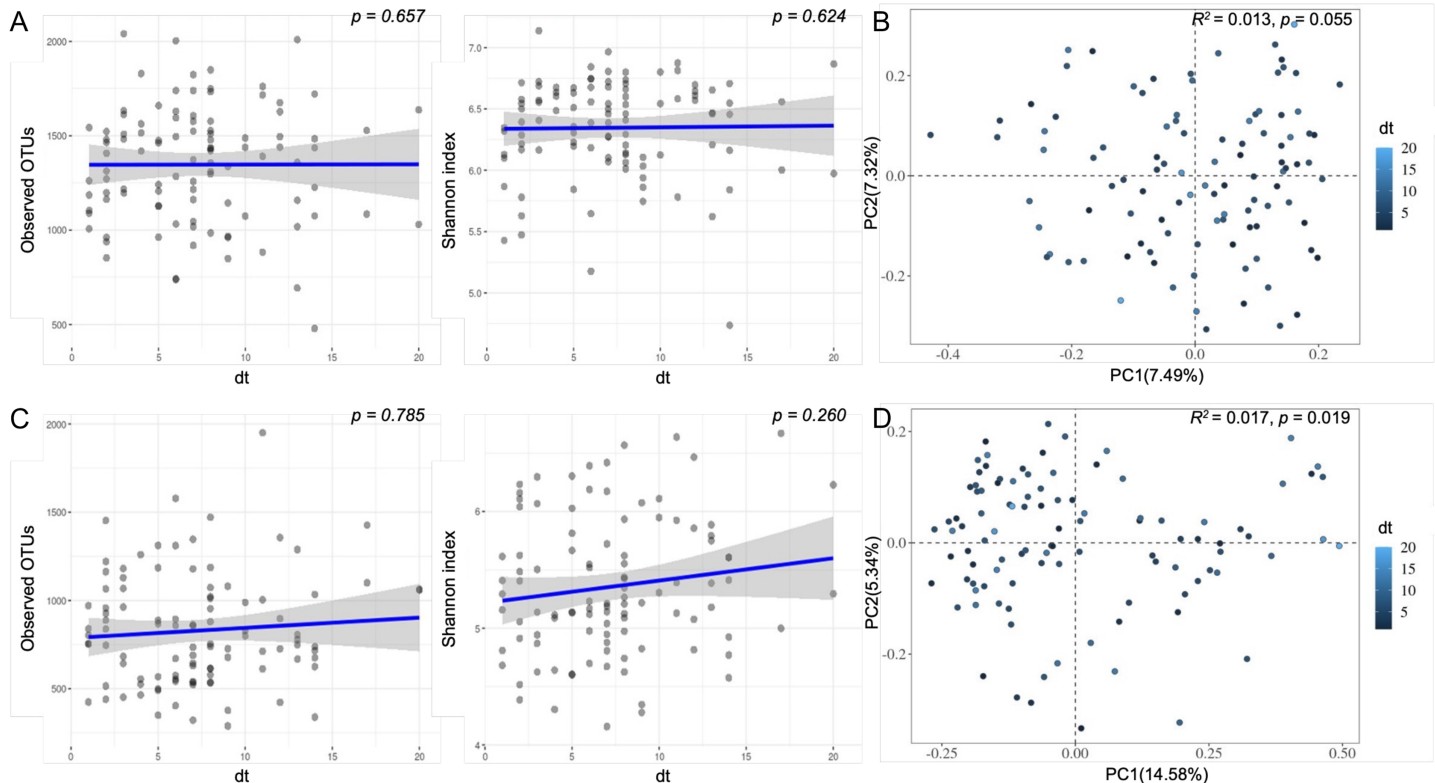

**Figure 2 Scatter plot of bacterial diversity of plaque and saliva samples correlated with dt.** (A and C) The observed and Shannon diversity of (A) plaque and (C) saliva samples correlated with dt using nonparametric Mann-Whitney U tests; (B and D) For beta diversity, Bray-Curtis distances were calculated, followed by PCoA. dt, decayed teeth.

*Fusobacteria*, and *Actinobacteria* were enriched in the plaque group. At the genus level (Fig. S3C), *Streptococcus* and *Haemophilus* were enriched in the saliva group, and *Neisseria, Leptotrichia, Fusobacterium, Corynebacterium, Actinomyces*, and *Veillonella* were rich in the plaque samples, and the differences were statistically significant ($p < 0.05$).

## The relationship between oral microbiota and dental caries in the overall population

To investigate the correlation between dental caries and the microbiome in this study, we stipulated that the severity of caries was expressed in terms of the number of decayed teeth (dt), and then we plotted the correlation of dt with flora.

In the plaque group, there was no significant difference in richness (observed OTUs, Kruskal-Wallis, $p = 0.657$; Fig. 2A), alpha diversity (Shannon index, Kruskal-Wallis, $p = 0.624$; Fig. 2A), or beta diversity (Bray-Curtis, PERMANOVA, $p = 0.055$; Fig. 2B). In the saliva group, there was no significant difference in richness (observed OTUs, Kruskal-Wallis, $p = 0.785$; Fig. 2C) or alpha diversity (Shannon index, Kruskal-Wallis, $p = 0.260$; Fig. 2C), but there was a significant difference in beta diversity by executing PCoA in terms of Bray-Curtis (PERMANOVA, $p = 0.019$; Fig. 2D).

Then, we used MaAsLin2 to identify the specific bacteria that were most associated with dental caries. Table 2 shows all the microorganisms associated with dt. In both groups,

| Table 2 The bacteria correlated with dt in the plaque and saliva groups. | | | | | | | |
|---|---|---|---|---|---|---|---|
| Group | Feature | P | | | S | | |
| | | $R^2$ | $p$ | $q$ | $R^2$ | $p$ | $q$ |
| **Both** | c__TM7.3 | 0.026 | 0.001 | 0.018 | 0.013 | 0.009 | 0.122 |
| | c__Clostridia | 0.026 | 0.027 | 0.191 | 0.017 | 0.026 | 0.123 |
| | p__Saccharibacteria | 0.026 | 0.001 | 0.012 | 0.013 | 0.009 | 0.078 |
| **P** | g__Acinetobacter | −0.006 | 0.01 | 0.246 | – | – | – |
| **S** | o__Actinomycetales | – | – | – | 0.039 | 0.019 | 0.185 |
| | o__Clostridiales | – | – | – | 0.017 | 0.026 | 0.185 |
| | c__Actinobacteria | – | – | – | 0.039 | 0.019 | 0.123 |
| | p__Actinobacteria | – | – | – | 0.039 | 0.019 | 0.087 |

**Note:**
P, supringival plaque; S, unstimulated saliva.

*Saccharibacteria* were the most associated bacteria. In the plaque group, *Acinetobacter* was the only microbiome related to dt, while *Actinomycetales* and *Clostridiales* were related to dt in the saliva group.

## Community profiling at different ages associated with caries

First, we distinguished plaque and saliva samples according to the participants' age, so we obtained four subgroups: preschoolers' plaque group, PP; schoolers' plaque group, PS; preschoolers' saliva group, SP; and schoolers' saliva group, SS. The results in Figs. 3A and 3C show that, in the P and S groups, there were no significant differences in richness and evenness, but there was a significant difference in β diversity between different ages (Bray–Curtis, PERMANOVA, $p < 0.05$, Figs. 3B and 3D).

In terms of the severity of caries, we plotted the correlation chart between different subgroups, as shown in Fig. 4. In the plaque group, only primary school children had a significant difference in beta diversity (Bray–Curtis, PERMANOVA, $p = 0.007$; Fig. 4D). In the saliva group, the preschool children had a significant difference in beta diversity (Bray–Curtis, PERMANOVA, $p = 0.032$; Fig. 4F). There were no significant differences in richness and evenness among the four groups.

## Community profiling in different sexes associated with caries

We further analyzed differences in the diversity of bacterial flora in plaque and saliva between children of different sexes. All samples were categorized by sampling site and gender, and four subgroups were obtained: male plaque group, PM; female plaque group, PF; male saliva group, SM; and female saliva group, SF. The results in Figs. 5A and 5C show that in the P and S groups, there were no significant differences in α and β diversity between the different sexes (Fig. 5, $p > 0.05$).

## The core microbiome in preschoolers and schoolers is associated with decayed teeth

Since there was no significant difference between females and males, we also intended to investigate the relationship between preschool and primary school children, so we used
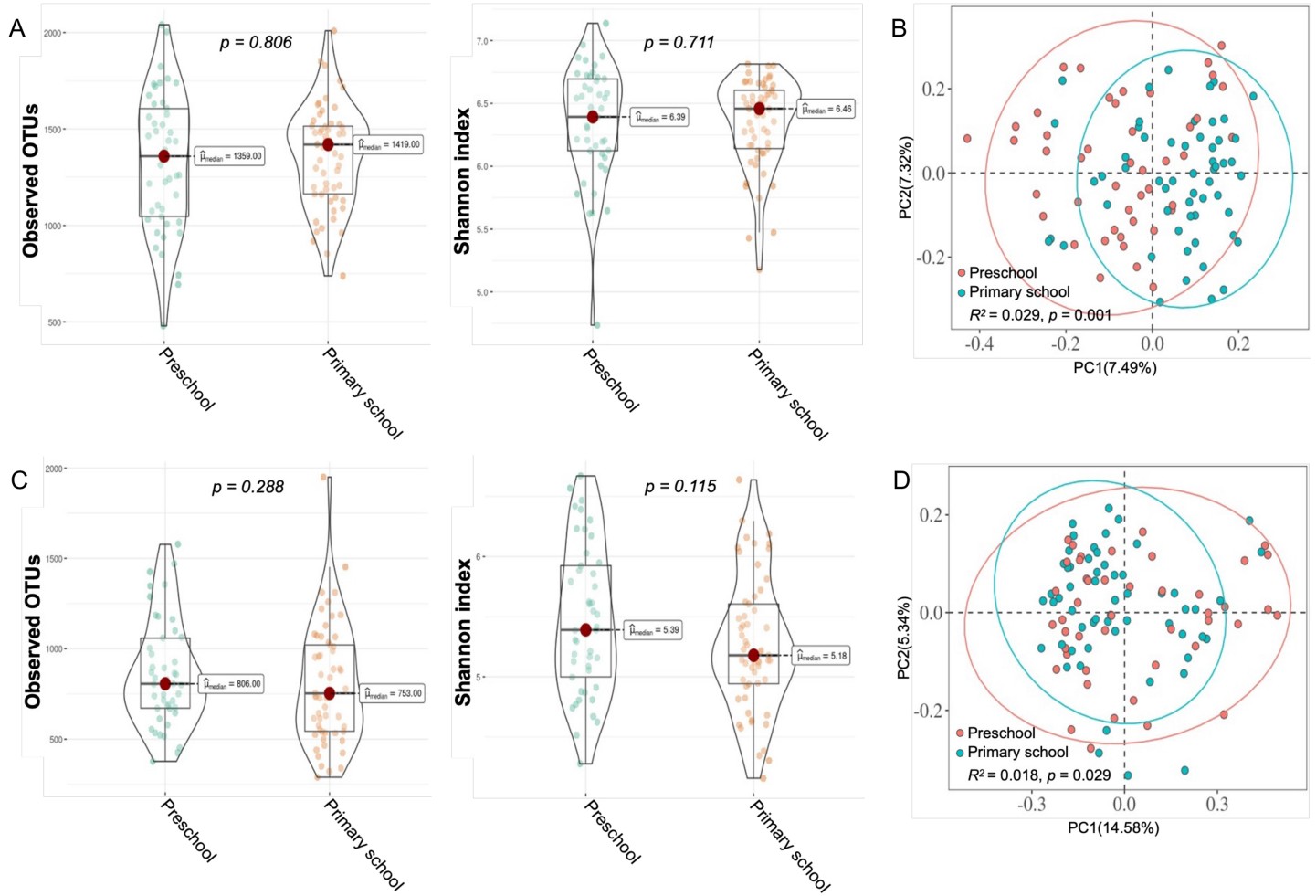

**Figure 3 Diversity of dental plaque and saliva samples from children within different ages.** (A and C) For alpha (within-sample) diversity, the observed OTUs and Shannon index were calculated according to (A) plaque samples and (C) saliva samples in both age groups. (Mann–Whitney U tests were used for group comparisons); (B and D) For beta (between-sample) diversity, Bray–Curtis distances were calculated, followed by PCoA. The plot shows the separation of (B) plaque samples and (D) saliva samples according to children's age.

MaAsLin2 to identify the specific microbiome. In plaque, *Saccharobacteria* was the only phylum associated with caries severity in both subgroups, and *Firmicutes* was the key bacteria in the PP group; *Clostridia, Bacilli* and *TM7.3* were the key bacteria in the PS group (Table 3). In the saliva group, no bacteria were the same in SP and SS; *Gemellaceae, Micrococcaceae*, and *Lachnospiraceae* were the key bacteria in the SP group; and *Capnocytophaga* and *Aggregatibacter* were the key bacteria in the SS group (Table 4). Differences between sexes are shown in the attached tables (Tables S1 and S2), and no bacteria were found in common between males and females.

According to the above results, we found that the age and severity of caries correlated significantly but that *Saccharibacteria* was the only type of bacteria associated with the severity of caries in both groups. The relative abundance of microbiome features was converted to units of log fold change, then we analyze the correlation between the

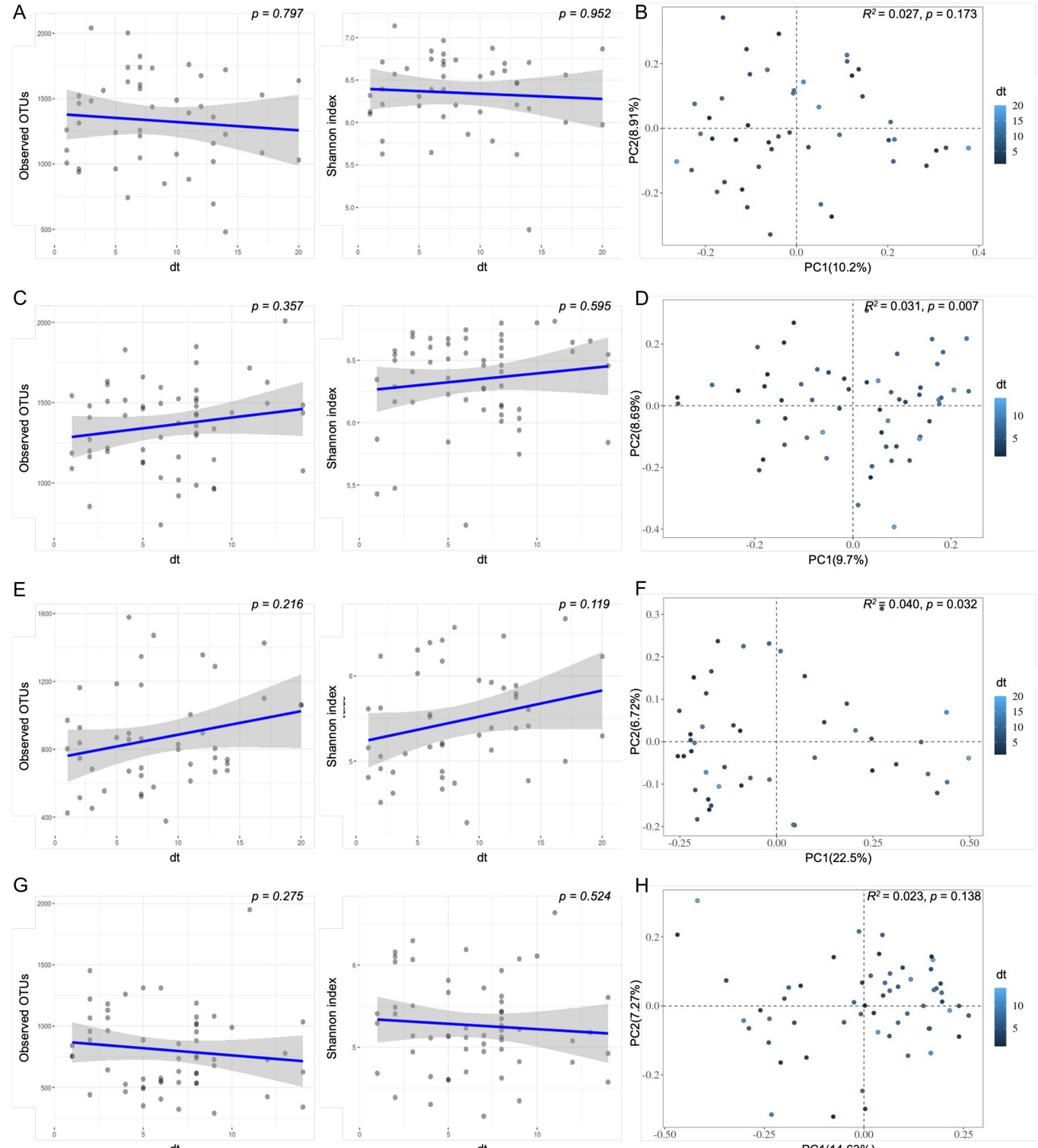

**Figure 4 Scatter plot of bacterial diversity of plaque and saliva samples within different ages.** The observed and Shannon diversity correlated with dt using nonparametric Mann–Whitney U tests; For beta diversity, Bray–Curtis distances were calculated, followed by PCoA. (A and B) Plaque Preschool; (C and D) Plaque Primary School; (E and F) Saliva Preschool; (G and H) Saliva Primary School. dt, decayed teeth.

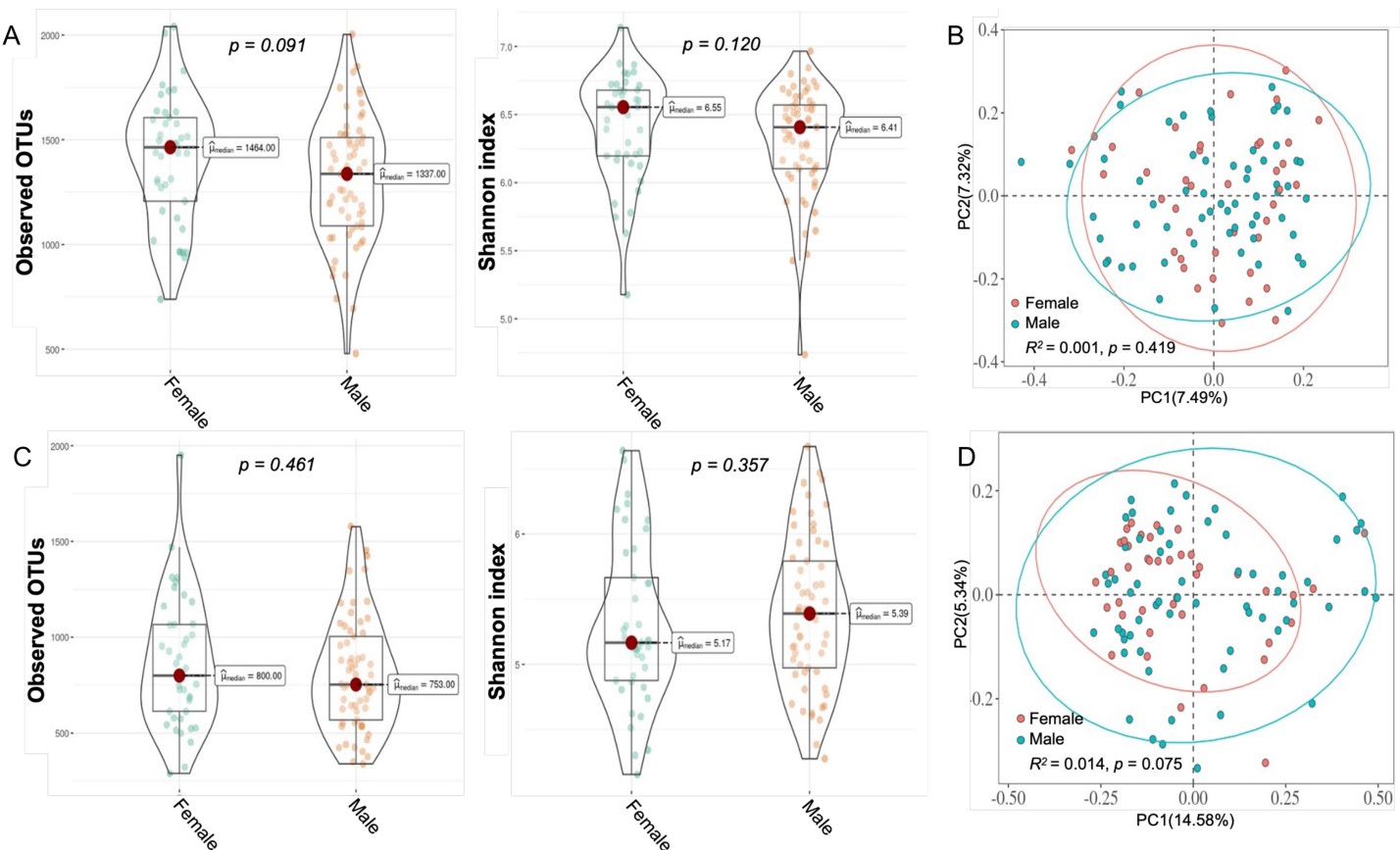

**Figure 5 Bacterial diversity of dental plaque and saliva samples from children within females and males.** (A and C) For alpha diversity (within-sample), the observed, Shannon diversity and richness measures were calculated according to (A) plaque samples and (C) salivary samples in both gender groups. Mann–Whitney U tests were used for group comparisons; (B and D) For beta (between-sample) diversity, Bray-Curtis distances were calculated, followed by PCoA. The plot shows the separation of (B) plaque samples and (D) saliva samples according to children's gender.

abundances of *Saccharibacteria* and other microbial genera. We found that *Saccharibacteria* abundance was associated with the abundance of many species at the genus level. Among these, the abundance of *Fusobacterium* demonstrated the highest positive correlation ($r = 0.586$), while the abundance of *Streptococcus* showed the highest negative correlation ($r = -0.593$) with the levels of *Saccharibacteria* (Fig. S4).

## DISCUSSION

The present study revealed the similarity and differences in the caries-associated microbiota by different oral sampling sites and among subpopulations. The results of this experiment showed that different sampling sites and different ages affected the identification of key microorganisms in caries. This study found that *Saccharibacteria* and age group at different positions is the only consistent bacterial classification, suggesting that it may have value in future research. In addition, the relationship between oral flora and caries may have population heterogeneity, so it is necessary to consider the specific population characteristics in future research and transformation.

Table 3 Plaque bacteria correlated with dt in the preschool and primary school groups.

| Group | Feature | PP | | | PS | | |
|---|---|---|---|---|---|---|---|
| | | $R^2$ | $p$ | $q$ | $R^2$ | $p$ | $q$ |
| Both | p__Saccharibacteria | 0.032 | 0.027 | 0.124 | 0.019 | 0.037 | 0.243 |
| PP | p__Firmicutes | 0.040 | 0.003 | 0.031 | – | – | – |
| PS | g__Paludibacter | – | – | – | −0.006 | 0.010 | 0.171 |
| | g__Porphyromonas | – | – | – | −0.015 | 0.012 | 0.171 |
| | g__Veillonella | – | – | – | 0.0420 | 0.009 | 0.171 |
| | g__Granulicatella_f__Carnobacteriaceae | – | – | – | −0.011 | 0.031 | 0.243 |
| | g__Moryella | – | – | – | 0.008 | 0.034 | 0.243 |
| | g__Actinobacillus | – | – | – | −0.012 | 0.032 | 0.243 |
| | f__Veillonellaceae | – | – | – | 0.044 | 0.006 | 0.111 |
| | f__Porphyromonadaceae | – | – | – | −0.014 | 0.015 | 0.177 |
| | f__Carnobacteriaceae | – | – | – | −0.011 | 0.031 | 0.248 |
| | o__Clostridiales | – | – | – | 0.048 | 0.003 | 0.073 |
| | c__Clostridia | – | – | – | 0.048 | 0.003 | 0.049 |
| | c__Bacilli | – | – | – | −0.035 | 0.047 | 0.189 |
| | c__TM7.3 | – | – | – | 0.019 | 0.037 | 0.189 |

Note:
PP, Preschoolers' supragingival plaque; PS, Schoolers' supragingival plaque.

Under normal physiological conditions, a dynamic balance is maintained between microorganisms and the host to jointly maintain the health of the host. However, the external environment or host factors can disrupt this balance, and some beneficial microorganisms become pathogenic, leading to dental caries, periodontal disease and other oral infectious diseases (*Gao et al., 2018*).

A previous study suggested that the overall composition and proportion of microbes differ among the different regions of the oral cavity (*Socransky & Manganiello, 1971*; *Gibbons & Houte, 1975*). Unlike the microbiome of other parts of the body, the oral microbiome is considered to be highly diverse (*Xiao, Fiscella & Gill, 2020*). Although dental plaque, saliva, and buccal mucosa are in close contact, they have different microbial communities (*Mark Welch, Ramírez-Puebla & Borisy, 2020*). *Hall et al. (2017)* found significant differences in the microbial communities present in supragingival plaque patches, saliva, and tongue samples from healthy subjects, suggesting the presence of site-specific oral microbial communities. The Human Microbiome Project (HMP) compared microbial diversity across five major body regions in 242 healthy individuals, and the results showed that supragingival plaque had a higher bacterial alpha diversity than the oral mucosa (*Human Microbiome Project Consortium, 2012*), which is consistent with the results of the present study.

Although saliva samples are easy to collect, distinctive taxonomic profiles between saliva and dental plaque have been observed in previous research (*Cui et al., 2021*; *de Jesus et al., 2021*). Our results showed that the microbial communities in supragingival plaque were significantly different in diversity from those in unstimulated saliva, and the microbial

**Table 4 The saliva bacteria correlated with dt in the preschool and primary school groups.**

| Group | Feature | SP | | | SS | | |
|---|---|---|---|---|---|---|---|
| | | $R^2$ | $p$ | $q$ | $R^2$ | $p$ | $q$ |
| SP | f__Gemellaceae | −0.030 | 0.006 | 0.111 | – | – | – |
| | f__Micrococcaceae | 0.028 | 0.049 | 0.240 | – | – | – |
| | f__Lachnospiraceae | 0.011 | 0.032 | 0.240 | – | – | – |
| | f__Cardiobacteriaceae | 0.008 | 0.041 | 0.240 | – | – | – |
| | o__Gemellales | −0.030 | 0.006 | 0.062 | – | – | – |
| | o__Actinomycetales | 0.058 | 0.034 | 0.162 | – | – | – |
| | o__Fusobacteriales | 0.042 | 0.037 | 0.162 | – | – | – |
| | o__Cardiobacteriales | 0.008 | 0.041 | 0.162 | – | – | – |
| | c__TM7.3 | 0.025 | 0.006 | 0.076 | – | – | – |
| | c__Actinobacteria | 0.058 | 0.034 | 0.120 | – | – | – |
| | c__Bacilli | −0.063 | 0.032 | 0.120 | – | – | – |
| | c__Fusobacteriia | 0.042 | 0.037 | 0.120 | – | – | – |
| | p__Saccharibacteria | 0.025 | 0.006 | 0.052 | – | – | – |
| | p__Actinobacteria | 0.058 | 0.034 | 0.111 | – | – | – |
| | p__Fusobacteria | 0.042 | 0.037 | 0.111 | – | – | – |
| SS | g__Capnocytophaga | – | – | – | −0.012 | 0.025 | 0.228 |
| | g__Aggregatibacter | – | – | – | −0.020 | 0.021 | 0.228 |
| | c__Flavobacteriia | – | – | – | −0.034 | 0.046 | 0.226 |
| | c__BD1.5 | – | – | – | −0.002 | 0.025 | 0.226 |
| | p__GN02 | – | – | – | −0.002 | 0.025 | 0.218 |
| | p__SR1 | – | – | – | −0.004 | 0.048 | 0.218 |

**Note:**
SP, Preschoolers' unstimulated saliva; SS, Schoolers' unstimulated saliva.

communities were more evenly distributed in supragingival plaque than in saliva. Plaques contain many microorganisms that closely adhere to the tooth surface and provide continuous nutrition for bacteria. The flow rate, viscosity, buffering capacity and remineralization capacity of saliva are important factors affecting caries (*Kim et al., 2021*), which regulate the progression and outcome of dental caries to some extent. By analyzing the correlation between the enriched groups and the number of carious teeth, we found that *Saccharibacteria* was the only bacterial group with an abundance that was most correlated with the severity of caries in both plaque and saliva, while other bacteria showed large differences, suggesting the heterogeneity of key bacteria across sites and the potential research value of *Saccharibacteria*.

*Saccharibacteria* (formerly known as *TM7*) are widely present in a variety of habitats, and their physiological and ecological roles and pathogenic properties remain unknown due to the difficulty of obtaining them by traditional culture methods (*McLean et al., 2020*). In previous studies, *Saccharibacteria* were found only in peri-implantitis pockets of aggressive periodontitis patients, leading scientists to suspect that this group of bacteria could cause periodontal disease (*Sousa et al., 2017*). *Baker, (2021)* found that an increase in

the levels of salivary immune markers (EGF, CSF2, IL13, *etc.*) has a similar trend with the increase in *Saccharibacteria* abundance.

Genomic analysis revealed that a *Saccharibacteria* representative (RAAC3) lacks nucleotide, lipid and amino acid biosynthetic pathways, suggesting that *Saccharibacteria* may be auxotrophic and metabolically dependent on other organisms (*Kantor et al., 2013*). "*Candidatus Saccharibacteria*" was proposed as the new phylum name for *Saccharibacteria* based on genomic analysis, which suggested that these bacteria, which show reduced genomes, primarily consume sugar compounds (*Bor et al., 2019*). In 2015, *He et al. (2015)* first cultivated a *TM7* phylotype (*TM7x*) from the human oral cavity. This microbe was cocultured with its bacterial host, *Actinomyces odontolyticus* subspecies *actinosynbacter*, XH001 (*Bedree et al., 2018*).

The results of this study suggested the presence of a relationship between *Saccharibacteria* and dental caries, but still could not explain the role of *Saccharibacteria* in dental caries. However, we observed that the abundance of *Saccharibacteria* was most negatively correlated with *Streptococcus* abundance, suggesting that there may be some interaction between them that needs further investigation. A number of studies have confirmed that among *Streptococcus spp*, *Streptococcus mutans* and *Streptococcus sobrinus* are the main causative agents of human dental caries (*Li, Wyllie & Jensen, 2021*; *Lemos et al., 2019*; *Philip, Suneja & Walsh, 2018*). *Saccharibacteria* may be a protective factor in this population. A recent study showed that *Saccharibacteria* suppresses gingival inflammation and bone loss in mice and could protect mammalian hosts from inflammatory damage induced by bacteria in the host (*Chipashvili et al., 2021*). The latest research shows that environmentally-derived *Saccharibacteria* acquired an arginine deiminase system during their evolution and colonization in the human oral cavity; this system can metabolize arginine to provide energy (ATP) and protects *Saccharibacteria* and its host bacteria against the acidic environment in oral cavity (*Tian et al., 2022*).

*Acinetobacter* was the key bacteria found in caries in the plaque group, while *Actinomycetales* and *Clostridiales* were the key bacteria found in caries in the saliva group. *Clostridiales*, a member of the "orange" complex in subgingival plaque, has traditionally been considered an important periodontal pathogen (*Socransky et al., 1998*). However, an increasing number of studies have noted its enrichment and predictive potential for ECC occurrence (*Zhu et al., 2018*; *Chen et al., 2021*). In the present study, the abundance of *Clostridiales* in saliva showed a significant positive correlation with dt, again highlighting its possible role in caries progression.

A large number of studies have shown that the diversity of the oral bacterial community in children with caries is lower than that in children without caries, and caries-related differential flora, such as *Streptococcus*, *Prevotella*, *Veillonella*, *Neisseria*, and *Rothia*, have been found (*de Jesus et al., 2021*, *2020*; *Wang et al., 2019*; *Baker et al., 2021*). This is similar in other inflammatory diseases in which the α diversity of the affected group is reduced (*Chen et al., 2020*).

Previous studies have suggested that the predominance of *S. mutans* may reduce community diversity (*Richards et al., 2017*). Some scholars have speculated that *S. mutans* may compete with other oral commensal bacteria to disrupt the microbial balance through

its strong capacity to produce acid, tolerance to acidic conditions and exopolysaccharide production (*Zhang et al., 2022*). In this study, due to the limitation of 16S sequencing at the species level, there was no in-depth analysis of which environments showed enrichment in *S. mutans*. However, it is certain that *Streptococcus* is the most abundant bacterium among different groups in the population of individuals with caries.

We found that the key bacteria associated with the severity of caries varied with the age of the children. In this study, *Saccharibacteria* was found to be the only bacterial category in supragingival plaque that was associated with caries severity in children of all ages, suggesting that there may be age differences in caries prevention targets of the oral microbiota. Some scholars have shown that, among bacterial genera with high relative abundance, only *TM7x* is strongly correlated with the risk of caries (*Kalpana et al., 2020*; *Baker et al., 2021*). This finding is consistent with the results of this experiment.

## CONCLUSIONS

In summary, our data indicated that, in a South China population, oral microbial signatures for dental caries show age and sex differences, but *Saccharibacteria* might be a consistent signal, which explains the inconsistency of results regarding the key bacteria reported in different studies. The identity of the key oral bacteria involved in caries remains unclear, and further population studies are needed to explore the potential influencing factors, which is in the beginning stages of research. However, this study is a cross-sectional survey, which can only reflect clues related to the etiology of dental caries and not the causal relationship. The relevant clues still need long-term and in-depth observation through cohort studies and omics studies, and the specific mechanism remains to be further explored. This study suggests possible target groups and locus heterogeneity and should, thus, be considered in future research and transformation according to the characteristics of the people in the corresponding design.

## ACKNOWLEDGEMENTS

We would like to thank Guangdong MagiGene Technology Co., LTD for sample sequencing and AJE for English language editing.

### Funding

This work was supported by grants from the President Foundation of Shenzhen Stomatology Hospital (Pingshan) of Southern Medical University (grant no. 2021A001), the Oral Infectious Disease Mechanism Research and Clinical Translation Application Innovation team of Guangdong Province of China (grant no. 2021KCXTD033) and the Chinese Stomatological Association Dental caries prevention and treatment capacity improvement program (grant no. CSA-ICP2022-03). The funders had no role in study design, data collection and analysis, decision to publish, or preparation of the manuscript.

## Grant Disclosures

The following grant information was disclosed by the authors:

President Foundation of Shenzhen Stomatology Hospital (Pingshan) of Southern Medical University: 2021A001.

Oral Infectious Disease Mechanism Research and Clinical Translation Application Innovation team of Guangdong Province of China: 2021KCXTD033.

Chinese Stomatological Association Dental caries prevention and treatment capacity improvement program: CSA-ICP2022-03.

## Competing Interests

The authors declare that they have no competing interests.

## Author Contributions

- Yang You conceived and designed the experiments, performed the experiments, analyzed the data, authored or reviewed drafts of the article, and approved the final draft.
- Meixiang Yin analyzed the data, prepared figures and/or tables, and approved the final draft.
- Xiao Zheng performed the experiments, prepared figures and/or tables, and approved the final draft.
- Qiuying Liang performed the experiments, prepared figures and/or tables, and approved the final draft.
- Hui Zhang analyzed the data, prepared figures and/or tables, and approved the final draft.
- Bu-Ling Wu conceived and designed the experiments, authored or reviewed drafts of the article, and approved the final draft.
- Wenan Xu conceived and designed the experiments, authored or reviewed drafts of the article, and approved the final draft.

## Human Ethics

The following information was supplied relating to ethical approvals (*i.e.*, approving body and any reference numbers):

The study was conducted in accordance with the Declaration of Helsinki, and approved by the Ethics Committee of Shenzhen Stomatology Hospital (Pingshan) of Southern Medical University (202201A).

## Data Availability

The 16S rRNA bacterial profiling data generated in this study are available at NCBI: PRJNA948117.

## Supplemental Information

Supplemental information for this article can be found online at http://dx.doi.org/10.7717/peerj.15605#supplemental-information.

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
