# Peer review of "Saccharibacteria (TM7), but not other bacterial taxa, are associated with childhood caries regardless of age in a South China population"

_PeerJ, doi:10.7717/peerj.15605_

## Round 0.1 · original submission · Major Revisions

I am sending it for major revisions. Please revise the manuscript and resubmit at the earliest.

Reviewer 1 ·

Basic reporting

The literature review part is not comprehensive. Given the recent discovery of TM7, there are only a few significant studies performed on this bacteria and they should be included in the manuscript, at the minimum discuss about in the manuscript and compare the findings between those papers and the author's current findings.

Experimental design

Line 195, "Our findings suggest that the bacterial community in the plaque is more evenly distributed than that in saliva". One major concern regarding the sample collection for unstimulated saliva. Patients were asked to place a cotton roll in their mouth for 1 min. Where in the oral cavity was the cotton roll placed? Is it consistent within all the patients? The cotton roll was in contact with tongue surface and the palate or side of the tooth? These could have all contributed to the discrepancy in the microbial samples. Cross contamination can easily happen if the cotton is in contact with the teeth or tongue.

Validity of the findings

The findings in the paper have been reported in other research. A few studies have established the increased in TM7 abundance in periodontal plaque sample and a reduction in Actinobacteria, the host bacteria for TM7. It’s surprising that the author did not discuss or reference these studies in their manuscript, eg PMID: 30319555, PMID: 34637779 etc).

TM7 is a obligate oral parasite, it cannot live on its own without a host bacterium. Depending on the oral environment, TM7 can kill its host bacterium. Since the author claimed that TM7 is associated with childhood caries, they should compare healthy vs caries in children to find out which bacteria phyla is missing or altered between the two groups, for example Actinomyces spp. which was shown to be the host for TM7 . The rise in TM7 could very well mean that a specific host bacterium is killed or enhanced to give rise to higher population of TM7

Additional comments

1. Line 44, include the full definition for of DMFT when first mentioned
2. Line 59, a balanced microbiota is the foundation of oral health,
3. Standardize the writing style, spacing before a reference or no spacing
4. Line 111, C does not need capitalization
5. Line 136, "replace" is a wrong word choice here
6. Line 169, t-test, t should be italicized
7. Line 171 p-value, q-value, p and q should be italicized
8. Line 191 p value should be italicized throughout the text
9. Line 206, please specify the supplementary figure in 3 that was used when discussing or referring to the figure, eg Fig. S3A, S3B, etc.

Reviewer 2 ·

Basic reporting

The paper is an in vitro study on the Saccharibacteria (TM7) association with childhood caries in a south China population.

The Authors made a great work in terms of methodology and the paper sounds scientific and well written.

However, some improvements are mandatory before acceptance.

The abstract is well written, complete and summary in its various aspects. The keywords are complete and appropriate.

The Introduction is well written, clear and complete in many respects. I think the authors have done a great job of analyzing a topic of this kind with such great clarity.
• “The relationship between bacterial flora and dental caries is complicated, and studies indicate that the population association characteristics of key oral bacteria for dental caries are still unclear.” I believe that a microbiological analysis indicative of the pathology or that can lead to a targeted drug therapy is increasingly important, avoiding the side effects of prolonged therapies with broad-spectrum drugs, both on the oral microbiota and at a systemic level, as I suggest to the authors to clarify in the introduction and suggest this research from this point of view updated, as indicated by: “Mahendra J, Mahendra L, Mugri MH, Sayed ME, Bhandi S, Alshahrani RT, Balaji TM, Varadarajan S, Tanneeru S, P ANR, Srinivasan S, Reda R, Testarelli L, Patil S. Role of Periodontal Bacteria, Viruses, and Placental mir155 in Chronic Periodontitis and Preeclampsia-A Genetic Microbiological Study. Curr Issues Mol Biol. 2021 Jul 29;43(2):831-844. doi: 10.3390/cimb43020060.”
• BASIC REPORTING is complete under several points of view, I believe that the authors have done an excellent job.

Experimental design

In the Materials and Methods section:
• The authors did a great job in the explication of all the variables identified and included in the study.

• EXPERIMENTAL DESIGN: the selection of the participants and the different inclusion criteria are of an excellent level, complete. The methodology followed is absolutely in line with the standard of these manuscripts.

Validity of the findings

Results are easy to understand and comprehensive. All the studied characteristics were reported in tables which are clear and concise.
• VALIDITY OF THE FINDINGS: are well represented, the methodology was of a high standard, and the different results were listed in a precise and high level manner, also considering the complexity of the study and the different variables examined.

Additional comments

In the Discussion:
• this section is complete and evaluates the outcome of different papers present in literature. The overall is comprehensive, concise and complete in its various aspects. This section too was organized and written in an absolutely positive way by the authors. Congratulations.
Conclusions are concise and clear.

Bibliography should be formatted respecting the journal’s requirements and no improper citations are evidenced.

Figures and labels are clear and easy to comprehend.

English is clear and easy to understand.

---

## Round 0.2 · accepted · Accept

The paper is accepted for publication.

Reviewer 1 ·

Basic reporting

The author has revised their manuscript according to previous suggestion and new literatures have been added. Thank you.

Experimental design

The concern regarding saliva collection has been clarified. Thank you.

Validity of the findings

"A number of studies have confirmed that among Streptococcus SPP, Streptococcus mutans and Streptococcus sobrinus are the main causative agents of human dental caries (Li, Wyllie & Jensen, 2021; Lemos et al., 2019; Philip, Suneja & Walsh, 2018). Saccharibacteria may be a protective factor in this population."

“Some scholars have shown that, among bacterial genera with high relative abundance, only TM7x is strongly correlated with the risk of caries (Kalpana et al., 2020; Baker et al., 2021).”

- current study suggested a negative correlation between TM7 and Streptococcus, where increase TM7 seems to be protective from caries. Perhaps the author can provide a few lines of speculation to justify their findings, maybe TM7 population increases as caries progresses and this in turn leads to reduction of S. mutans to slow down caries progression?

Additional comments

line 335, revise Streptococcus SPP to Streptococcus spp, no capital letters needed.
line 359, S. mutans should be italicized

Reviewer 2 ·

Basic reporting

The article is well written, the Authors have substantially improved the quality of the manuscript.

Experimental design

The methodology has been improved in the suggested points.

Validity of the findings

The results are interesting and of good quality.